# Incremental Variational Sparse Gaussian Process Regression

**Ching-An Cheng**
Institute for Robotics and Intelligent Machines
Georgia Institute of Technology
Atlanta, GA 30332
cacheng@gatech.edu

**Byron Boots**
Institute for Robotics and Intelligent Machines
Georgia Institute of Technology
Atlanta, GA 30332
bboots@cc.gatech.edu

## Abstract

Recent work on scaling up Gaussian process regression (GPR) to large datasets has primarily focused on sparse GPR, which leverages a small set of basis functions to approximate the full Gaussian process during inference. However, the majority of these approaches are batch methods that operate on the entire training dataset at once, precluding the use of datasets that are streaming or too large to fit into memory. Although previous work has considered incrementally solving variational sparse GPR, most algorithms fail to update the basis functions and therefore perform suboptimally. We propose a novel incremental learning algorithm for variational sparse GPR based on stochastic mirror ascent of probability densities in reproducing kernel Hilbert space. This new formulation allows our algorithm to update basis functions online in accordance with the manifold structure of probability densities for fast convergence. We conduct several experiments and show that our proposed approach achieves better empirical performance in terms of prediction error than the recent state-of-the-art incremental solutions to variational sparse GPR.

## 1 Introduction

Gaussian processes (GPs) are nonparametric statistical models widely used for probabilistic reasoning about functions. Gaussian process regression (GPR) can be used to infer the distribution of a latent function $f$ from data. The merit of GPR is that it finds the *maximum a posteriori* estimate of the function while providing the profile of the remaining uncertainty. However, GPR also has drawbacks: like most nonparametric learning techniques the time and space complexity of GPR scale polynomially with the amount of training data. Given $N$ observations, inference of GPR involves inverting an $N \times N$ covariance matrix which requires $O(N^3)$ operations and $O(N^2)$ storage. Therefore, GPR for large $N$ is infeasible in practice.

Sparse Gaussian process regression is a pragmatic solution that trades accuracy against computational complexity. Instead of parameterizing the posterior using *all* $N$ observations, the idea is to approximate the full GP using the statistics of finite $M \ll N$ function values and leverage the induced low-rank structure to reduce the complexity to $O(M^2 N + M^3)$ and the memory to $O(M^2)$. Often sparse GPRs are expressed in terms of the distribution of $f(\tilde{x}_i)$, where $\tilde{X} = \{\tilde{x}_i \in \mathcal{X}\}_{i=1}^{M}$ are called *inducing points* or *pseudo-inputs* [21, 23, 18, 26]. A more general representation leverages the information about the *inducing function* $(L_i f)(\tilde{x}_i)$ defined by indirect measurement of $f$ through a bounded linear operator $L_i$ (e.g. integral) to more compactly capture the full GP [27, 8]. In this work, we embrace the general notion of inducing functions, which trivially includes $f(\tilde{x}_i)$ by choosing $L_i$ to be identity. With abuse of notation, we reuse the term inducing points $\tilde{X}$ to denote the parameters that define the inducing functions.

Learning a sparse GP representation in regression can be summarized as inference of the hyperparameters, the inducing points, and the statistics of inducing functions. One approach to learning is to treat all of the parameters as hyperparameters and find the solution that maximizes the marginal likelihood [21, 23, 18]. An alternative approach is to view the inducing points and the statistics of inducing functions as variational parameters of a class of full GPs, to approximate the true posterior of $f$, and solve the problem via variational inference, which has been shown robust to over-fitting [26, 1].

All of the above methods are designed for the batch setting, where all of the data is collected in advance and used at once. However, if the training dataset is extremely large or the data are streaming and encountered in sequence, we may want to *incrementally* update the approximate posterior of the latent function $f$. Early work by Csató and Opper [6] proposed an online version of GPR, which greedily performs moment matching of the true posterior given *one* sample instead of the posterior of *all* samples. More recently, several attempts have been made to modify variational batch algorithms to incremental algorithms for learning sparse GPs [1, 9, 10]. Most of these methods rely on the fact that variational sparse GPR with fixed inducing points and hyperparameters is equivalent to inference of the conjugate exponential family: Hensman et al. [9] propose a stochastic approximation of the variational sparse GPR problem [26] based on stochastic natural gradient ascent [11]; Hoang et al. [10] generalizes this approach to the case with general Gaussian process priors. Unlike the original variational algorithm for sparse GPR [26], which finds the optimal inducing points and hyperparameters, these algorithms only update the statistics of the inducing functions $f_{\tilde{X}}$.

In this paper, we propose an incremental learning algorithm for variational sparse GPR, which we denote as *iVSGPR*. Leveraging the dual formulation of variational sparse GPR in reproducing kernel Hilbert space (RKHS), *iVSGPR* performs stochastic mirror ascent in the space of probability densities [17] to update the approximate posterior of $f$, and stochastic gradient ascent to update the hyperparameters. Stochastic mirror ascent, similar to stochastic natural gradient ascent, considers the manifold structure of probability functions and therefore converges faster than the naive gradient approach. In each iteration, *iVSGPR* solves a variational sparse GPR problem of the size of a minibatch. As a result, *iVSGPR* has constant complexity per iteration and can learn all the hyperparameters, the inducing points, and the associated statistics online.

## 2  Background

In this section, we provide a brief summary of Gaussian process regression and sparse Gaussian process regression for efficient inference before proceeding to introduce our incremental algorithm for variational sparse Gaussian process regression in Section 3.

### 2.1  Gaussian Processes Regression

Let $\mathcal{F}$ be a family of real-valued continuous functions $f : \mathcal{X} \mapsto \mathbb{R}$. A GP is a distribution of functions $f$ in $\mathcal{F}$ such that, for any finite set $X \subset \mathcal{X}$, $\{f(x)|x \in X\}$ is Gaussian distributed $\mathcal{N}(f(x)|m(x), k(x, x'))$: for any $x, x' \in \mathcal{X}$, $m(x)$ and $k(x, x')$ represent the mean of $f(x)$ and the covariance between $f(x)$ and $f(x')$, respectively. In shorthand, we write $f \sim \mathcal{GP}(m, k)$.

The mean $m(x)$ and the covariance $k(x, x')$ (the kernel function) are often parametrized by a set of hyperparameters which encode our prior belief of the unknown function $f$. In this work, for simplicity, we assume that $m(x) = 0$ and the kernel can be parameterized as $k(x, x') = \rho^2 g_s(x, x')$, where $g_s(x, x')$ is a positive definite kernel, $\rho^2$ is a scaling factor and $s$ denotes other hyperparameters [20].

The objective of GPR is to infer the posterior probability of the function $f$ given data $\mathcal{D} = \{(x_i, y_i)\}_{i=1}^N$. In learning, the function value $f(x_i)$ is treated as a latent variable and the observation $y_i = f(x_i) + \epsilon_i$ is modeled as the function corrupted by *i.i.d.* noise $\epsilon_i \sim \mathcal{N}(\epsilon|0, \sigma^2)$. Let $X = \{x_i\}_{i=1}^N$. The posterior probability distribution $p(f|y)$ can be compactly summarized as $\mathcal{GP}(m_{|\mathcal{D}}, k_{|\mathcal{D}})$:

$$m_{|\mathcal{D}}(x) = k_{x,X}(K_X + \sigma^2 I)^{-1} y \tag{1}$$

$$k_{|\mathcal{D}}(x, x') = k_{x,x'} - k_{x,X}(K_X + \sigma^2 I)^{-1} k_{X,x'} \tag{2}$$

where $y = (y_i)_{i=1}^N \in \mathbb{R}^N$, $k_{x,X} \in \mathbb{R}^{1 \times N}$ denotes the vector of the cross-covariance between $x$ and $X$, and $K_X \in \mathbb{R}^{N \times N}$ denotes the empirical covariance matrix of the training set. The hyperparameters

$\theta := (s, \rho, \sigma)$ in the GP are learned by maximizing the log-likelihood of the observation $y$

$$\max_{\theta} \log p(y) = \max_{\theta} \log \mathcal{N}(y|0, K_X + \sigma^2 I). \tag{3}$$

## 2.2 Sparse Gaussian Processes Regression

A straightforward approach to sparse GPR is to approximate the GP prior of interest with a degenerate GP [21, 23, 18]. Formally, for any $x_i, x_j \in \mathcal{X}$, it assumes that

$$f(x_i) \perp y_i | f_{\tilde{X}}, \quad f(x_i) \perp f(x_j) | f_{\tilde{X}}, \tag{4}$$

where $f_{\tilde{X}}$ denotes $((L_i f)(\tilde{x}_i))_{i=1}^{M}$ and $\perp$ denotes probabilistic independence between two random variables. That is, the original empirical covariance matrix $K_X$ is replaced by a rank-$M$ approximation $\hat{K}_X := K_{X,\tilde{X}} K_{\tilde{X}}^{-1} K_{\tilde{X},X}$, where $K_{\tilde{X}}$ is the covariance of $f_{\tilde{X}}$ and $K_{X,\tilde{X}} \in \mathbb{R}^{N \times M}$ is the cross-covariance between $f_X$ and $f_{\tilde{X}}$. Let $\Lambda \in \mathbb{R}^{N \times N}$ be diagonal. The inducing points $\tilde{X}$ are treated as hyperparameters and can be found by jointly maximizing the log-likelihood with $\theta$

$$\max_{\theta, \tilde{X}} \log \mathcal{N}(y|0, \hat{K}_X + \sigma^2 I + \Lambda), \tag{5}$$

Several approaches to sparse GPR can be viewed as special cases of this problem [18]: the Deterministic Training Conditional (DTC) [21] approximation sets $\Lambda$ as zero. To heal the degeneracy in $p(f_X)$, the Fully Independent Training Conditional (FITC) approximation [23] includes heteroscedastic noise, setting $\Lambda = diag(K_X - \hat{K}_X)$. As a result, their sum $\Lambda + \hat{K}_X$ matches the true covariance $K_X$ in the diagonal term. This general maximum likelihood scheme for finding the inducing points is adopted with variations in [24, 27, 8, 2]. A major drawback of all of these approaches is that they can over-fit due to the high degrees-of-freedom $\tilde{X}$ in the prior parametrization [26].

Variational sparse GPR can alternatively be formulated to approximate the posterior of the latent function by a full GP parameterized by the inducing points and the statistics of inducing functions [1, 26]. Specifically, Titsias [26] proposes to use

$$q(f_X, f_{\tilde{X}}) = p(f_X | f_{\tilde{X}}) q(f_{\tilde{X}}) \tag{6}$$

to approximate $p(f_X, f_{\tilde{X}} | y)$, where $q(f_{\tilde{X}}) = \mathcal{N}(f_{\tilde{X}} | \tilde{m}, \tilde{S})$ is the Gaussian approximation of $p(f_{\tilde{X}} | y)$ and $p(f_X | f_{\tilde{X}}) = \mathcal{N}(f_X | K_{X,\tilde{X}} K_{\tilde{X}}^{-1} f_{\tilde{X}}, K_X - \hat{K}_X)$ is the conditional probability in the full GP. The novelty here is that $q(f_X, f_{\tilde{X}})$, despite parametrization by finite parameters, is still a full GP, which, unlike its predecessor [21], can be infinite-dimensional.

The inference problem of variational sparse GPR is solved by minimizing the KL-divergence $\text{KL}[q(f_X, f_{\tilde{X}}) \| p(f_X, f_{\tilde{X}} | y)]$. In practice, the minimization problem is transformed into the maximization of the lower bound of the log-likelihood [26]:

$$
\begin{aligned}
\max_{\theta} \log p(y) &\geq \max_{\theta, \tilde{X}, \tilde{m}, \tilde{S}} \int q(f_X, f_{\tilde{X}}) \log \frac{p(y|f_X) p(f_X | f_{\tilde{X}}) p(f_{\tilde{X}})}{q(f_X, f_{\tilde{X}})} \mathrm{d}f_X \mathrm{d}f_{\tilde{X}} \\
&= \max_{\theta, \tilde{X}, \tilde{m}, \tilde{S}} \int p(f_X | f_{\tilde{X}}) q(f_{\tilde{X}}) \log \frac{p(y|f_X) p(f_{\tilde{X}})}{q(f_{\tilde{X}})} \mathrm{d}f_X \mathrm{d}f_{\tilde{X}} \\
&= \max_{\theta, \tilde{X}} \log \mathcal{N}(y|0, \hat{K}_X + \sigma^2 I) - \frac{1}{2\sigma^2} \text{Tr}(K_X - \hat{K}_X).
\end{aligned} \tag{7}
$$

The last equality results from exact maximization over $\tilde{m}$ and $\tilde{S}$; for treatment of non-conjugate likelihoods, see [22]. We note that $q(f_{\tilde{X}})$ is a function of $\tilde{m}$ and $\tilde{S}$, whereas $p(f_{\tilde{X}})$ and $p(f_X | f_{\tilde{X}})$ are functions of $\tilde{X}$. As a result, $\tilde{X}$ become variational parameters that can be optimized without over-fitting. Compared with (5), the variational approach in (7) regularizes the learning with penalty $\text{Tr}(K_X - \hat{K}_X)$ and therefore exhibits better generalization performance. Several subsequent works employ similar strategies: Alvarez et al. [3] adopt the same variational approach in the multi-output regression setting with scaled basis functions, and Abdel-Gawad et al. [1] use expectation propagation to solve for the approximate posterior under the same factorization.

# 3 Incremental Variational Sparse Gaussian Process Regression

Despite leveraging sparsity, the batch solution to the variational objective in (7) requires $O(M^2N)$ operations and access to all of the training data during each optimization step [26], which means that learning from large datasets is still infeasible. Recently, several attempts have been made to *incrementally* solve the variational sparse GPR problem in order to learn better models from large datasets [1, 9, 10]. The key idea is to rewrite (7) explicitly into the sum of individual observations:

$$\max_{\theta, \tilde{X}, \tilde{m}, \tilde{S}} \int p(f_X|f_{\tilde{X}}) q(f_{\tilde{X}}) \log \frac{p(y|f_X)p(f_{\tilde{X}})}{q(f_{\tilde{X}})} \mathrm{d}f_X \mathrm{d}f_{\tilde{X}}$$

$$= \max_{\theta, \tilde{X}, \tilde{m}, \tilde{S}} \int q(f_{\tilde{X}}) \left( \sum_{i=1}^{N} \mathbb{E}_{p(f_{x_i}|f_{\tilde{X}})} [\log p(y_i|f_{x_i})] + \log \frac{p(f_{\tilde{X}})}{q(f_{\tilde{X}})} \right) \mathrm{d}f_{\tilde{X}}. \tag{8}$$

The objective function in (8), with fixed $\tilde{X}$, is identical to the problem of stochastic variational inference [11] of conjugate exponential families. Hensman et al. [9] exploit this idea to incrementally update the statistics $\tilde{m}$ and $\tilde{S}$ via stochastic *natural* gradient ascent,[1] which, at the $t$th iteration, takes the direction derived from the limit of maximizing (8) subject to $\mathrm{KL}^{sym}(q_t(f_{\tilde{X}})||q_{t-1}(f_{\tilde{X}})) < \epsilon$ as $\epsilon \to 0$. Natural gradient ascent considers the manifold structure of probability distribution derived from KL divergence and is known to be Fisher efficient [4]. Although the optimal inducing points $\tilde{X}$, like the statistics $\tilde{m}$ and $\tilde{S}$, should be updated given new observations, it is difficult to design natural gradient ascent for learning the inducing points $\tilde{X}$ online. Because $p(f_X|f_{\tilde{X}})$ in (8) depends on all the observations, evaluating the divergence with respect to $p(f_X|f_{\tilde{X}})q(f_{\tilde{X}})$ over iterations becomes infeasible.

We propose a novel approach to incremental variational sparse GPR, *iVSGPR*, that works by reformulating (7) in its RKHS dual form. This avoids the issue of the posterior approximation $p(f_X|f_{\tilde{X}})q(f_{\tilde{X}})$ referring to all observations. As a result, we can perform stochastic approximation of (7) while monitoring the KL divergence between the posterior approximates due to the change of $\tilde{m}$, $\tilde{S}$, and $\tilde{X}$ across iterations. Specifically, we use stochastic mirror ascent [17] in the space of probability densities in RKHS, which was recently proven to be as efficient as stochastic natural gradient ascent [19]. In each iteration, *iVSGPR* solves a subproblem of fractional Bayesian inference, which we show can be formulated into a standard variational sparse GPR of the size of a minibatch in $O(M^2N_m + M^3)$ operations, where $N_m$ is the size of a minibatch.

## 3.1 Dual Representations of Gaussian Processes in RKHS

An RKHS $\mathcal{H}$ is a Hilbert space of functions satisfying the reproducing property: $\exists k_x \in \mathcal{H}$ such that $\forall f \in \mathcal{H}, f(x) = \langle f, k_x \rangle_{\mathcal{H}}$. In general, $\mathcal{H}$ can be infinite-dimensional and uniformly approximate continuous functions on a compact set [16]. To simplify the notation we write $k_x^T f$ for $\langle f, k_x \rangle_{\mathcal{H}}$, and $f^T L g$ for $\langle f, Lg \rangle$, where $f, g \in \mathcal{H}$ and $L : \mathcal{H} \mapsto \mathcal{H}$, even if $\mathcal{H}$ is infinite-dimensional.

A Gaussian process $\mathcal{GP}(m, k)$ has a dual representation in an RKHS $\mathcal{H}$ [12]: there exists $\mu \in \mathcal{H}$ and a positive semi-definite linear operator $\Sigma : \mathcal{H} \mapsto \mathcal{H}$ such that for any $x, x' \in \mathcal{X}, \exists \phi_x, \phi_{x'} \in \mathcal{H}$,

$$m(x) = \rho \phi_x^T \mu, \quad k(x, x') = \rho^2 \phi_x^T \Sigma \phi_{x'}. \tag{9}$$

That is, the mean function has a realization $\mu$ in $\mathcal{H}$, which is defined by the reproducing kernel $\tilde{k}(x, x') = \rho^2 \phi_x^T \phi_{x'}$; the covariance function can be equivalently represented by a linear operator $\Sigma$. In shorthand, with abuse of notation, we write $\mathcal{N}(f|\mu, \Sigma)$.[2] Note that we do not assume the samples from $\mathcal{GP}(m, k)$ are in $\mathcal{H}$. In the following, without loss of generality, we assume the GP prior considered in regression has $\mu = 0$ and $\Sigma = I$. That is, $m(x) = 0$ and $k(x, x') = \rho^2 \phi_x^T \phi_{x'}$.

### 3.1.1 Subspace Parametrization of the Approximate Posterior

The full GP posterior approximation $p(f_X|f_{\tilde{X}})q(f_{\tilde{X}})$ in (7) can be written equivalently in a *subspace parametrization* using $\{\psi_{\tilde{x}_i} \in \mathcal{H} | \tilde{x}_i \in \tilde{X}\}_{i=1}^{M}$:

$$\tilde{\mu} = \Psi_{\tilde{X}} a, \quad \tilde{\Sigma} = I + \Psi_{\tilde{X}} A \Psi_{\tilde{X}}^T, \tag{10}$$

where $a \in \mathbb{R}^M$, $A \in R^{M \times M}$ such that $\tilde{\Sigma} \succeq 0$, and $\Psi_{\tilde{X}} : \mathbb{R}^M \mapsto \mathcal{H}$ is defined as $\Psi_{\tilde{X}} a = \sum_{i=1}^M a_i \psi_{\tilde{x}_i}$. Suppose $q(f_{\tilde{X}}) = \mathcal{N}(f_{\tilde{X}} | \tilde{m}, \tilde{S})$ and define $\psi_{\tilde{x}_i}$ to satisfy $\Psi_{\tilde{X}}^T \tilde{\mu} = \tilde{m}$. By (10), $\tilde{m} = K_{\tilde{X}} a$ and $\tilde{S} = K_{\tilde{X}} + K_{\tilde{X}} A K_{\tilde{X}}$, which implies the relationship

$$a = K_{\tilde{X}}^{-1} \tilde{m}, \quad A = K_{\tilde{X}}^{-1} \tilde{S} K_{\tilde{X}}^{-1} - K_{\tilde{X}}^{-1}, \tag{11}$$

where the covariances related to the inducing functions are defined as $K_{\tilde{X}} = \Psi_{\tilde{X}}^T \Psi_{\tilde{X}}$ and $K_{X,\tilde{X}} = \rho \Phi_X^T \Psi_{\tilde{X}}$. The sparse structure results in $f(x) \sim \mathcal{GP}(k_{x,\tilde{X}} K_{\tilde{X}}^{-1} \tilde{m}, k_{x,x} + k_{x,\tilde{X}}(K_{\tilde{X}}^{-1} \tilde{S} K_{\tilde{X}}^{-1} - K_{\tilde{X}}^{-1}) k_{\tilde{X},x})$, which is the same as $\int p(f(x)|f_{\tilde{X}}) q(f_{\tilde{X}}) \mathrm{d} f_{\tilde{X}}$, the posterior GP found in (7), where $k_{x,x} = k(x,x)$ and $k_{x,\tilde{X}} = \rho \phi_x^T \Psi_{\tilde{X}}$. We note that the scaling factor $\rho$ is associated with the evaluation of $f(x)$, not the inducing functions $f_{\tilde{X}}$. In addition, we distinguish the hyperparameter $s$ (e.g. length scale) that controls the measurement basis $\phi_x$ from the parameters in inducing points $\tilde{X}$.

A subspace parametrization corresponds to a full GP if $\tilde{\Sigma} \succ 0$. More precisely, because (10) is completely determined by the statistics $\tilde{m}$, $\tilde{S}$, and the inducing points $\tilde{X}$, the family of subspace parametrized GPs lie on a nonlinear submanifold in the space of all GPs (the degenerate GP in (4) is a special case if we allow $I$ in (10) to be ignored).

### 3.1.2 Sparse Gaussian Processes Regression in RKHS

We now reformulate the variational inference problem (7) in RKHS[3]. Following the previous section, the sparse GP structure on the posterior approximate $q(f_X, f_{\tilde{X}})$ in (6) has a corresponding dual representation in RKHS $q(f) = \mathcal{N}(f|\tilde{\mu}, \tilde{\Sigma})$. Specially, $q(f)$ and $q(f_X, f_{\tilde{X}})$ are related as follows:

$$q(f) \propto p(f_X | f_{\tilde{X}}) q(f_{\tilde{X}}) |K_{\tilde{X}}|^{1/2} |K_X - \hat{K}_X|^{1/2}, \tag{12}$$

in which the determinant is due to the change of measure. The equality (12) allows us to rewrite (7) in terms of $q(f)$ simply as

$$\max_{q(f)} \mathcal{L}(q(f)) = \max_{q(f)} \int q(f) \log \frac{p(y|f)p(f)}{q(f)} \mathrm{d}f, \tag{13}$$

or equivalently as $\min_{q(f)} \mathrm{KL}[q(f)||p(f|y)]$. That is, the heuristically motivated variational problem (7) is indeed minimizing a proper KL-divergence between two Gaussian *measures*. A similar justification on (7) is given rigorously in [14] in terms of KL-divergence minimization between Gaussian *processes*, which can be viewed as a dual of our approach. Due to space limitations, the proofs of (12) and the equivalence between (7) and (13) can be found in the Appendix.

The benefit of the formulation of (13) is that in its sampling form,

$$\max_{q(f)} \int q(f) \left( \sum_{i=1}^N \log p(y_i|f) + \log \frac{p(f)}{q(f)} \right) \mathrm{d}f, \tag{14}$$

the approximate posterior $q(f)$ nicely summarizes all the variational parameters $\tilde{X}$, $\tilde{m}$, and $\tilde{S}$ without referring to the samples as in $p(f_X|f_{\tilde{X}}) q(f_{\tilde{X}})$. Therefore, the KL-divergence of $q(f)$ across iterations can be used to regulate online learning.

### 3.2 Incremental Learning

Stochastic mirror ascent [17] considers (non-)Euclidean structure on variables induced by a Bregman divergence (or prox-function) [5] in convex optimization. We apply it to solve the variational inference problem in (14), because (14) is convex in the space of probabilities [17]. Here, we ignore the dependency of $q(f)$ on $f$ for simplicity. At the $t$th iteration, stochastic mirror ascent solves the subproblem

$$q_{t+1} = \arg \max_q \gamma_t \int \hat{\partial} \mathcal{L}(q_t, y_t) q(f) \mathrm{d}f - \mathrm{KL}[q||q_t], \tag{15}$$

where $\gamma_t$ is the step size, $\hat{\partial}\mathcal{L}(q_t, y_t)$ is the sampled subgradient of $\mathcal{L}$ with respect to $q$ when the observation is $(x_t, y_t)$. The algorithm converges in $O(t^{-1/2})$ if (15) is solved within numerical error $\epsilon_t$ such that $\sum \epsilon_t \sim O(\sum \gamma_t^2)$ [7].

The subproblem (15) is actually equivalent to sparse variational GP regression with a general Gaussian prior. By definition of $\mathcal{L}(q)$ in (14), (15) can be derived as

$$q_{t+1} = \arg\max_q \gamma_t \int q(f) \left( N \log p(y_t|f) + \log \frac{p(f)}{q_t(f)} \right) \mathrm{d}f - \mathrm{KL}[q||q_t]$$

$$= \arg\max_q \int q(f) \log \frac{p(y_t|f)^{N\gamma_t} p(f)^{\gamma_t} q_t^{1-\gamma_t}(f)}{q(f)} \mathrm{d}f. \tag{16}$$

This equation is equivalent to (13) but with the prior modified to $p(f)^{\gamma_t} q_t(f)^{1-\gamma_t}$ and the likelihood modified to $p(y_i|f)^{N\gamma_t}$. Because $p(f)$ is an isotropic zero-mean Gaussian, $p(f)^{\gamma_t} q_t(f)^{1-\gamma_t}$ has the subspace parametrization expressed in the same basis functions as $q_t$. Suppose $q_t$ has mean $\tilde{\mu}_t$ and precision $\tilde{\Sigma}_t^{-1}$. Then $p(f)^{\gamma_t} q_t(f)^{1-\gamma_t}$ is equivalent to $\mathcal{N}(f|\hat{\mu}, \hat{\Sigma})$ up to a constant factor, where $\hat{\mu}_t = (1-\gamma_t)\hat{\Sigma}_t \tilde{\Sigma}_t^{-1} \tilde{\mu}_t$ and $\hat{\Sigma}_t^{-1} = (1-\gamma_t)\tilde{\Sigma}_t^{-1} + \gamma_t I$. By (10), $\tilde{\Sigma}_t^{-1} = I - \Psi_{\tilde{X}}(A_t^{-1} + K_{\tilde{X}})^{-1}\Psi_{\tilde{X}}$ for some $A_t$, and therefore $\hat{\Sigma}_t^{-1} = I - (1-\gamma_t)\Psi_{\tilde{X}}(A_t^{-1} + K_{\tilde{X}})^{-1}\Psi_{\tilde{X}}$, which is expressed in the same basis. In addition, by (12), (16) can be further written in the form of (7) and therefore solved by a standard sparse variational GPR program with modified $\tilde{m}$ and $\tilde{S}$ (Please see Appendix for details).

Although we derived the equations for a single observation, minibatchs can be used with the same convergence rate and reduced variance by changing the factor $p(y_t|f)^{N\gamma_t}$ to $\prod_{i=1}^{N_m} p(y_{t_i}|f)^{\frac{N\gamma_t}{N_m}}$. The hyperparameters $\theta = (s, \rho, \sigma)$ in the GP can be updated along with the variational parameters using stochastic gradient ascent along the gradient of $\int q_t(f) \log \frac{p(y_t|f)p(f)}{q_t(f)} \mathrm{d}f$.

### 3.3  Related Work

The subproblem (16) is equivalent to first performing stochastic natural gradient ascent [11] of $q(f)$ in (14) and then projecting the distribution back to the low-dimensional manifold specified by the subspace parametrization. At the $t$th iteration, define $q_t'(f) := p(y_t|f)^{N\gamma_t} p(f)^{\gamma_t} q_t(f)^{1-\gamma_t}$. Because a GP can be viewed as Gaussian measure in an infinite-dimensional RKHS, $q_t'(f)$ (16) can be viewed as the result of taking natural stochastic gradient ascent with step size $\gamma_t$ from $q_t(f)$. Then (16) becomes $\min_q \mathrm{KL}[q||q_t']$ in order to project $q_t'$ back to subspace parametrization specified by $M$ basis functions. Therefore, (16) can also be viewed as performing stochastic natural gradient ascent with a KL divergence projection. From this perspective, we can see that if $\tilde{X}$, which controls the inducing functions, are fixed in the subproblem (16), *iVSGPR* degenerates to the algorithm of Hensman et al. [9].

Recently, several researches have considered the manifold structure induced by KL divergence in Bayesian inference [7, 25, 13]. Theis and Hoffman [25] use trust regions to mitigate the sensitivity of stochastic variational inference to choices of hyperparameters and initialization. Let $\xi_t$ be the size of the trust region. At the $t$th iteration, it solves the objective $\max_q \mathcal{L}(q) - \xi_t \mathrm{KL}[q||q_t]$, which is the same as subproblem (16) if applied to (14). The difference is that in (16) $\gamma_t$ is a decaying step sequence in stochastic mirror ascent, whereas $\xi_t$ is manually selected. A similar formulation also appears in [13], where the part of $\mathcal{L}(q)$ non-convex to the variational parameters is linearized. Dai et al. [7] use particles or a kernel density estimator to approximate the posterior of $\tilde{X}$ in the setting with low-rank GP prior. By contrast, we follow Titsias's variational approach [26] to adopt a full GP as the approximate posterior, and therefore avoid the difficulties in estimating the posterior of $\tilde{X}$ and focus on the approximate posterior $q(f)$ related to the function values.

The stochastic mirror ascent framework sheds light on the convergence condition of the algorithm. As pointed out in Dai et al. [7], the subproblem (15) can be solved up to $\epsilon_t$ accuracy as long as $\sum \epsilon_t$ is order $O(\sum \gamma_t^2)$, where $\gamma_t \sim O(1/\sqrt{t})$ [17]. Also, Khan et al. [13] solves a linearized approximation of (15) in each step and reports satisfactory empirical results. Although variational sparse GPR (16) is a nonconvex optimization in $\tilde{X}$ and is often solved by nonlinear conjugate gradient ascent, empirically the objective function increases most significantly in the first few iterations. Therefore, based on the results in [7], we argue that in online learning (16) can be solved approximately by performing a small fixed number of line searches.

# 4 Experiments

We compare our method *iVSGPR* with *VSGPR*$_\text{svi}$ the state-of-the-art variational sparse GPR method based on stochastic variational inference [9], in which *i.i.d.* data are sampled from the training dataset to update the models. We consider a zero-mean GP prior generated by a squared-exponential with automatic relevance determination (SE-ARD) kernel [20] $k(x, x') = \rho^2 \prod_{d=1}^{D} \exp(\frac{-(x_d - x'_d)^2}{2s_d^2})$, where $s_d > 0$ is the length scale of dimension $d$ and $D$ is the dimensionality of the input. For the inducing functions, we modified the multi-scale kernel in [27] to

$$\psi_x^T \psi_{x'} = \prod_{i=d}^{D} \left( \frac{2l_{x,d}l_{x',d}}{l_{x,d}^2 + l_{x',d}^2} \right)^{1/2} \exp\left( -\sum_{d=1}^{D} \frac{\|x_d - x'_d\|^2}{l_{x,d}^2 + l_{x',d}^2} \right), \tag{17}$$

where $l_{x,d}$ is the length-scale parameter. The definition (17) includes the SE-ARD kernel as a special case, which can be recovered by identifying $\psi_x = \phi_x$ and $(l_{x,d})_{d=1}^{D} = (s_d)_{d=1}^{D}$, and hence their cross covariance can be computed.

In the following experiments, we set the number inducing functions to 512. All models were initialized with the same hyperparameters and inducing points: the hyperparameters were selected as the optimal ones in the batch variational sparse GPR [26] trained on subset of the training dataset of size 2048; the inducing points were initialized as random samples from the first minibatch. We chose the learning rate to be $\gamma_t = (1 + \sqrt{t})^{-1}$, for stochastic mirror ascent to update the posterior approximation; the learning rate for the stochastic gradient ascent to update the hyperparameters is set to $10^{-4}\gamma_t$. We evaluate the models in terms of the normalized mean squared error (nMSE) on a held-out test set after 500 iterations.

We performed experiments on three real-world robotic datasets datasets, *kin40k*[4], *SARCOS*[5], *KUKA*[6], and three variations of *iVSGPR*: *iVSGPR*$_5$, *iVSGPR*$_{10}$, and *iVSGPR*$_\text{ada}$.[7] For the *kin40k* and *SARCOS* datasets, we also implemented *VSGPR*$_\text{svi}^*$, which uses stochastic variational inference to update $\tilde{m}$ and $\tilde{S}$ but fixes hyperparameters and inducing points as the solution to the batch variational sparse GPR [26] with *all* of the training data. Because *VSGPR*$_\text{svi}^*$ reflects the perfect scenario of performing stochastic approximation under the selected learning rate, we consider it as the optimal goal we want to approach.

The experimental results of *kin40k* and *SARCOS* are summarized in Table 1a. In general, the adaptive scheme *iVSGPR*$_\text{ada}$ performs the best, but we observe that even performing a small fixed number of iterations ( *iVSGPR*$_5$, *iVSGPR*$_{10}$) results in performance that is close to, if not better than *VSGPR*$_\text{svi}^*$. Possible explanations are that the change of objective function in gradient-based algorithms is dominant in the first few iterations and that the found inducing points and hyper-parameters have finite numerical resolution in batch optimization. For example, Figure 1a shows the change of test error over iterations in learning joint 2 of *SARCOS* dataset. For all methods, the convergence rate improves with a larger minibatch. In addition, from Figure 1b, we observe that the required number of steps *iVSGPR*$_\text{ada}$ needed to solve (16) decays with the number of iterations; only a small number line searches is required after the first few iterations.

Table 1b and Table 1c show the experimental results on two larger datasets. In the experiments, we mixed the offline and online partitions in the original *KUKA* dataset and then split 90% into training and 10% into testing datasets in order to create an online *i.i.d.* streaming scenario. We did not compare to *VSGPR*$_\text{svi}^*$ on these datasets, since computing the inducing points and hyperparameters in batch is infeasible. As above, *iVSGPR*$_\text{ada}$ stands out from other models, closely followed by *iVSGPR*$_{10}$. We found that the difference between *VSGPR*$_\text{svi}$ and *iVSGPR*s is much greater on these larger real-world benchmarks.

Auxiliary experimental results illustrating convergence for all experiments summarized in Tables 1a, 1b, and 1c are included in the Appendix.

| | $VSGPR_{svi}$ | $iVSGPR_5$ | $iVSGPR_{10}$ | $iVSGPR_{ada}$ | $VSGPR^*_{svi}$ |
|---|---|---|---|---|---|
| kin40k | 0.0959 | 0.0648 | 0.0608 | **0.0607** | 0.0535 |
| SARCOS $J_1$ | 0.0247 | 0.0228 | 0.0214 | **0.0210** | 0.0208 |
| SARCOS $J_2$ | 0.0193 | 0.0176 | 0.0159 | **0.0156** | 0.0156 |
| SARCOS $J_3$ | 0.0125 | 0.0112 | 0.0104 | **0.0103** | 0.0104 |
| SARCOS $J_4$ | 0.0048 | 0.0044 | 0.0040 | **0.0038** | 0.0039 |
| SARCOS $J_5$ | 0.0267 | 0.0243 | 0.0229 | **0.0226** | 0.0230 |
| SARCOS $J_6$ | 0.0300 | 0.0259 | 0.0235 | **0.0229** | 0.0230 |
| SARCOS $J_7$ | 0.0101 | 0.0090 | 0.0082 | **0.0081** | 0.0101 |

(a) *kin40k* and *SARCOS*

| | $VSGPR_{svi}$ | $iVSGPR_5$ | $iVSGPR_{10}$ | $iVSGPR_{ada}$ |
|---|---|---|---|---|
| $J_1$ | 0.1699 | 0.1455 | 0.1257 | **0.1176** |
| $J_2$ | 0.1530 | 0.1305 | 0.1221 | **0.1138** |
| $J_3$ | 0.1873 | 0.1554 | 0.1403 | **0.1252** |
| $J_4$ | 0.1376 | 0.1216 | 0.1151 | **0.1108** |
| $J_5$ | 0.1955 | 0.1668 | 0.1487 | **0.1398** |
| $J_6$ | 0.1766 | 0.1645 | 0.1573 | **0.1506** |
| $J_7$ | 0.1374 | 0.1357 | 0.1342 | **0.1333** |

(b) KUKA1

| | $VSGPR_{svi}$ | $iVSGPR_5$ | $iVSGPR_{10}$ | $iVSGPR_{ada}$ |
|---|---|---|---|---|
| $J_1$ | 0.1737 | 0.1452 | 0.1284 | **0.1214** |
| $J_2$ | 0.1517 | 0.1312 | 0.1183 | **0.1081** |
| $J_3$ | 0.2108 | 0.1818 | 0.1652 | **0.1544** |
| $J_4$ | 0.1357 | 0.1171 | 0.1104 | **0.1046** |
| $J_5$ | 0.2082 | 0.1846 | 0.1697 | **0.1598** |
| $J_6$ | 0.1925 | 0.1890 | 0.1855 | **0.1809** |
| $J_7$ | 0.1329 | 0.1309 | 0.1287 | **0.1275** |

(c) KUKA2

Table 1: Testing error (nMSE) after 500 iterations. $N_m = 2048$; $J_i$ denotes the $i$th joint.

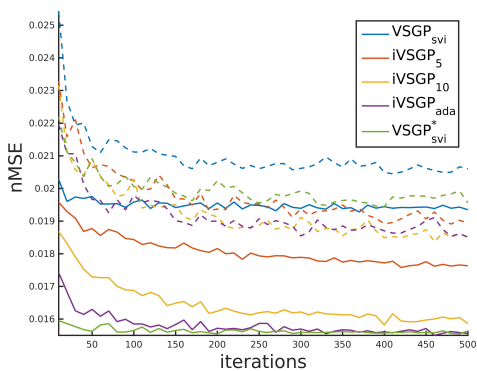

(a) Test error

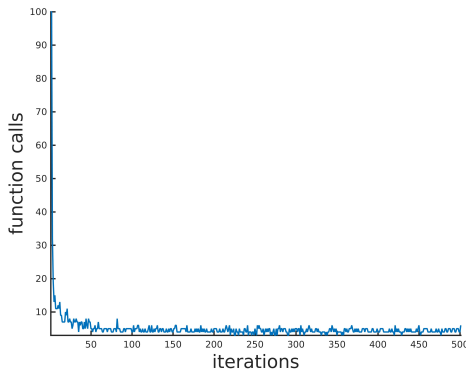

(b) Functions calls of *iVSGPR*$_{ada}$

Figure 1: Online learning results of *SARCOS* joint 2. (a) nMSE evaluated on the held out test set; the dash lines and the solid lines denote the results with $N_m = 512$ and $N_m = 2048$, respectively. (b) Number of function calls used by *iVSGPR*$_{ada}$ in solving (16) (A maximum of 100 calls is imposed )

## 5   Conclusion

We propose a stochastic approximation of variational sparse GPR [26], *iVSGPR*. By reformulating the variational inference in RKHS, the update of the statistics of the inducing functions and the inducing points can be unified as stochastic mirror ascent on probability densities to consider the manifold structure. In our experiments, *iVSGPR* shows better performance than the direct adoption of stochastic variational inference to solve variational sparse GPs. As *iVSGPR* executes a fixed number of operations for each minibatch, it is suitable for applications where training data is abundant, e.g. sensory data in robotics. In future work, we are interested in applying *iVSGPR* to extensions of sparse Gaussian processes such as GP-LVMs and dynamical system modeling.

## Footnotes

[1]Although $\tilde{X}$ was fixed in their experiments, it can potentially be updated by stochastic gradient ascent.

[2]Because a GP can be infinite-dimensional, it cannot define a density but only a Gaussian measure. The notation $\mathcal{N}(f|\mu, \Sigma)$ is used to indicate that the Gaussian measure can be defined, equivalently, by $\mu$ and $\Sigma$.

[3]Here we assume the set $X$ is finite and countable. This assumption suffices in learning and allows us to restrict $\mathcal{H}$ be the finite-dimensional span of $\Phi_X$. Rigorously, for infinite-dimensional $\mathcal{H}$, the equivalence can be written in terms of Radon–Nikodym derivative between $q(f)df$ and normal Gaussian measure, where $q(f)df$ denotes a Gaussian measure that has an RKHS representation given as $q(f)$

[4]*kin40k*: 10000 training data, 30000 testing data, 8 attributes [23]

[5]*SARCOS*: 44484 training data, 4449 testing data, 28 attributes. http://www.gaussianprocess.org/gpml/data/

[6]*KUKA1&KUKA2*: 17560 offline data, 180360 online data, 28 attributes. [15]

[7]The number in the subscript denotes the number of function calls allowed in nonlinear conjugate gradient descent [20] to solve subproblems (16) and $ada$ denotes (16) is solved until the relative function change is less than $10^{-5}$.

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
