[Supplementary Material · supplement.pdf]

# A Proofs

## A.1 Proof of (12)

Consider the subspace parametrization for the density $q(f) = \mathcal{N}(f|\tilde{\mu}, \tilde{\Sigma})$ of $f \in \mathcal{H}$ with

$$\tilde{\mu} = \Psi_{\tilde{X}} K_{\tilde{X}}^{-1} \tilde{m}$$
$$\tilde{\Sigma} = I + \Psi_{\tilde{X}} (K_{\tilde{X}}^{-1} \tilde{S} K_{\tilde{X}}^{-1} - K_{\tilde{X}}^{-1}) \Psi_{\tilde{X}}^T.$$

Decompose $f = f_{\parallel} + f_{\perp}$, where $f_{\parallel} = \Psi_{\tilde{X}} K_{\tilde{X}}^{-1} f_{\tilde{X}}$ and $f_{\perp}$ satisfies $N_{\tilde{X}} f_{\perp} = f_{\perp}$, with respect to the null-space projection $N_{\tilde{X}} = I - P_{\tilde{X}}$, where $P_{\tilde{X}} = \Psi_{\tilde{X}} K_{\tilde{X}}^{-1} \Psi_{\tilde{X}}^T$. Further, consider $b$ satisfying $f_{\perp} = \Phi_X b$, which implies $\Phi_X^T P_{\tilde{X}} \Phi_X b = \hat{K}_X b = 0$. That is, $b = \hat{N}b$, where $\hat{N}$ is the null space of $\hat{K}_X$. By construction, since

$$f_X - K_{X,\tilde{X}} K_{\tilde{X}}^{-1} f_{\tilde{X}} = \Phi_X^T (I - \Psi_{\tilde{X}} K_{\tilde{X}}^{-1} \Psi_{\tilde{X}}^T) f$$
$$= \Phi_X^T f_{\perp} = \Phi_X^T N_{\tilde{X}} \Phi_X \hat{N} b$$
$$= (K_X - \hat{K}_X) \hat{N} b$$

it follows that

$$-\log q(f) = \frac{1}{2} \log |\tilde{\Sigma}| + \frac{1}{2} (f - \tilde{\mu})^T \tilde{\Sigma}^{-1} (f - \tilde{\mu}) + \text{const.}$$

$$= \frac{1}{2} \log \frac{|\tilde{S}|}{|K_{\tilde{X}}|} + \frac{1}{2} (f - \tilde{\mu})^T (I - \Psi_{\tilde{X}} \left( K_{\tilde{X}}^{-1} - \tilde{S}^{-1} \right) \Psi_{\tilde{X}}^T)(f - \tilde{\mu}) + \text{const.}$$

$$= \frac{1}{2} \log \frac{|\tilde{S}|}{|K_{\tilde{X}}|} + \frac{1}{2} (f - \tilde{\mu})^T (N_{\tilde{X}} + \Psi_{\tilde{X}} \tilde{S}^{-1} \Psi_{\tilde{X}}^T)(f - \tilde{\mu}) + \text{const.}$$

$$= \frac{1}{2} \log \frac{|\tilde{S}|}{|K_{\tilde{X}}|} + \frac{1}{2} f_{\perp}^T N_{\tilde{X}} f_{\perp} + \frac{1}{2} (f_{\tilde{X}} - \tilde{m})^T K_{\tilde{X}}^{-1} \Psi_{\tilde{X}}^T \Psi_{\tilde{X}} \tilde{S}^{-1} \Psi_{\tilde{X}}^T \Psi_{\tilde{X}} K_{\tilde{X}}^{-1} (f_{\tilde{X}} - \tilde{m}) + \text{const.}$$

$$= \frac{1}{2} \log \frac{|\tilde{S}|}{|K_{\tilde{X}}|} + \frac{1}{2} b^T \hat{N} (K_X - \hat{K}_X) \hat{N} b + \frac{1}{2} (f_{\tilde{X}} - \tilde{m})^T \tilde{S}^{-1} (f_{\tilde{X}} - \tilde{m}) + \text{const.}$$

$$= \frac{1}{2} \log \frac{|\tilde{S}|}{|K_{\tilde{X}}|} + \frac{1}{2} (f_X - K_{X,\tilde{X}} K_{\tilde{X}}^{-1} f_{\tilde{X}})^T (K_X - \hat{K}_X)^+ (f_X - K_{X,\tilde{X}} K_{\tilde{X}}^{-1} f_{\tilde{X}})$$
$$+ \frac{1}{2} (f_{\tilde{X}} - \tilde{m})^T \tilde{S}^{-1} (f_{\tilde{X}} - \tilde{m}) + \text{const.}$$

$$= \frac{1}{2} \log \frac{1}{|K_{\tilde{X}}||K_X - \hat{K}_X|} - \log p(f_X|f_{\tilde{X}}) - \log q(f_{\tilde{X}}) + \text{const.}$$

where we used the identities

$$|\tilde{\Sigma}| = |I||(\tilde{S} - K_{\tilde{X}})^{-1} + K_{\tilde{X}}^{-1}||\tilde{S} - K_{\tilde{X}}|$$
$$= |(K_{\tilde{X}} - K_{\tilde{X}} \tilde{S}^{-1} K_{\tilde{X}})^{-1}||\tilde{S} - K_{\tilde{X}}|$$
$$= \frac{|\tilde{S} - K_{\tilde{X}}|}{|K_{\tilde{X}} - K_{\tilde{X}} \tilde{S}^{-1} K_{\tilde{X}}|}$$
$$= \frac{|\tilde{S} - K_{\tilde{X}}|}{|K_{\tilde{X}} \tilde{S}^{-1}||\tilde{S} - K_{\tilde{X}}|}$$
$$= \frac{|\tilde{S}|}{|K_{\tilde{X}}|}$$

and

$$\tilde{\Sigma}^{-1} = \left(I + \Psi_{\tilde{X}} K_{\tilde{X}}^{-1}(\tilde{S} - K_{\tilde{X}})K_{\tilde{X}}^{-1}\Psi_{\tilde{X}}^T\right)^{-1}$$
$$= (I - \Psi_{\tilde{X}} K_{\tilde{X}}^{-1}\left((\tilde{S} - K_{\tilde{X}})^{-1} + K_{\tilde{X}}^{-1}\right)^{-1} K_{\tilde{X}}^{-1}\Psi_{\tilde{X}}^T)$$
$$= (I - \Psi_{\tilde{X}} K_{\tilde{X}}^{-1}\left(K_{\tilde{X}} - K_{\tilde{X}}(K_{\tilde{X}} + \tilde{S} - K_{\tilde{X}})^{-1}K_{\tilde{X}}\right) K_{\tilde{X}}^{-1}\Psi_{\tilde{X}}^T)$$
$$= I - \Psi_{\tilde{X}} \left(K_{\tilde{X}}^{-1} - \tilde{S}^{-1}\right)\Psi_{\tilde{X}}^T.$$

Thus

$$q(f) \propto p(f_X|f_{\tilde{X}})q(f_{\tilde{X}})|K_{\tilde{X}}|^{1/2}|K_X - \hat{K}_X|^{1/2}$$

## A.2 Proof of the equivalence between (7) and (13)

Using (12), since

$$f_{\parallel}^T f_{\parallel} = f_{\tilde{X}}^T K_{\tilde{X}}^{-1} f_{\tilde{X}}$$

and

$$f_{\perp}^T f_{\perp} = b\hat{N}\Phi_X^T N_{\tilde{X}}\Phi_X \hat{N}b = (f_X - K_{X,\tilde{X}} K_{\tilde{X}}^{-1} f_{\tilde{X}})^T (K_X - \hat{K}_X)^+ (f_X - K_{X,\tilde{X}} K_{\tilde{X}}^{-1} f_{\tilde{X}})$$

we can rewrite (13) as

$$\int q(f)\log\frac{p(y|f)p(f)}{q(f)}\mathrm{d}f$$
$$\cong \int p(f_X|f_{\tilde{X}})q(f_{\tilde{X}})|K_{\tilde{X}}|^{1/2}|K_X - \hat{K}_X|^{1/2}\log\frac{p(y|f)p(f_X|f_{\tilde{X}})p(f_{\tilde{X}})|K_{\tilde{X}}|^{1/2}|K_X - \hat{K}_X|^{1/2}}{p(f_X|f_{\tilde{X}})q(f_{\tilde{X}})|K_{\tilde{X}}|^{1/2}|K_X - \hat{K}_X|^{1/2}}\mathrm{d}f$$
$$= \int p(f_X|f_{\tilde{X}})q(f_{\tilde{X}})\log\frac{p(y|f)p(f_X|f_{\tilde{X}})p(f_{\tilde{X}})}{p(f_X|f_{\tilde{X}})q(f_{\tilde{X}})}|K_{\tilde{X}}|^{1/2}\mathrm{d}f_{\parallel}|K_X - \hat{K}_X|^{1/2}\mathrm{d}f_{\perp}$$
$$= \int q(f_X, f_{\tilde{X}})\log\frac{p(y|f_X)p(f_{\tilde{X}})}{q(f_X, f_{\tilde{X}})}\mathrm{d}f_X\mathrm{d}f_{\tilde{X}}$$

where $\cong$ denotes the equivalence up to constants.

## A.3 Solution to subproblem (16)

Consider the objective function

$$\int q(f)\log\frac{p(y_t|f)^{N\gamma_t}p(f)^{\gamma_t}q_t(f)^{1-\gamma_t}}{q(f)}\mathrm{d}f$$

The modified likelihood term is

$$\log p(y_t|f)^{N\gamma_t} = \log\mathcal{N}(y_t|\phi_x^T f, \frac{\sigma^2}{N\gamma_t}) + \text{const.}$$

Suppose $q_t$ has mean $\tilde{\mu}_t$ and precision $\tilde{\Sigma}_t^{-1}$, where $\tilde{\Sigma}$ is subspace parametrized $\tilde{\Sigma}_t = I + \Psi_{\tilde{X}} A_t \Psi_{\tilde{X}}^T$ with $A_t = \tilde{S}_t^{-1} - K_{\tilde{X}}^{-1}$. Then $\tilde{\Sigma}_t^{-1} = I - \Psi_{\tilde{X}}(A_t^{-1} + K_{\tilde{X}})^{-1}\Psi_{\tilde{X}}$, and the natural parameters in the modified prior $p(f)^{\gamma_t}q_t(f)^{1-\gamma_t} \propto \mathcal{N}(f|\hat{\mu}_t, \hat{\Sigma}_t)$ can be written as

$$\hat{\Sigma}_t^{-1}\hat{\mu}_t = \gamma_t\Sigma^{-1}\mu + (1 - \gamma_t)\tilde{\Sigma}_t^{-1}\mu_t = (1 - \gamma_t)\tilde{\Sigma}_t^{-1}\mu_t$$
$$\hat{\Sigma}_t^{-1} = \gamma_t\Sigma^{-1} + (1 - \gamma_t)\tilde{\Sigma}_t^{-1} = I - (1 - \gamma_t)\Psi_{\tilde{X}}(A_t^{-1} + K_{\tilde{X}})^{-1}\Psi_{\tilde{X}}$$

In implementation, it means $p(f)^{\gamma_t}q_t(f)^{1-\gamma_t} \propto p(f_X|f_{\tilde{X}})q(f_{\tilde{X}}|\hat{m}, \hat{S})$, where $\hat{m}, \hat{S}$ can be identified as below:

$$\hat{\Sigma}_t^{-1} = I - (1 - \gamma_t)\Psi_{\tilde{X}}\left(K_{\tilde{X}}^{-1} - \tilde{S}_t^{-1}\right)\Psi_{\tilde{X}}^T$$

$$= I - \Psi_{\tilde{X}}\left(K_{\tilde{X}}^{-1} - K_{\tilde{X}}^{-1} + (1 - \gamma_t)K_{\tilde{X}}^{-1} - (1 - \gamma_t)\tilde{S}_t^{-1}\right)\Psi_{\tilde{X}}^T$$

$$= I - \Psi_{\tilde{X}}\left(K_{\tilde{X}}^{-1} - ((1 - \gamma_t)\tilde{S}_t^{-1} + \gamma_t K_{\tilde{X}}^{-1})\right)\Psi_{\tilde{X}}^T$$

$$= I - \Psi_{\tilde{X}}\left(K_{\tilde{X}}^{-1} - \hat{S}_t^{-1}\right)\Psi_{\tilde{X}}^T$$

where we define
$$\hat{S}_t^{-1} := (1 - \gamma_t)\tilde{S}_t^{-1} + \gamma_t K_{\tilde{X}}^{-1}$$

That is, subspace parametrization can be expressed with $\hat{S}$

$$\hat{\Sigma}_t = I - \Psi_{\tilde{X}}K_{\tilde{X}}^{-1}\left(K_{\tilde{X}} - \hat{S}_t\right)K_{\tilde{X}}^{-1}\Psi_{\tilde{X}}^T$$

For the mean,

$$\hat{\mu}_t = \hat{\Sigma}_t\left((1 - \gamma_t)\tilde{\Sigma}_t^{-1}\tilde{\mu}\right)$$

$$= (1 - \gamma_t)\hat{\Sigma}_t\tilde{\Sigma}_t^{-1}\tilde{\mu}$$

$$= (1 - \gamma_t)(I - \Psi_{\tilde{X}}K_{\tilde{X}}^{-1}\left(K_{\tilde{X}} - \hat{S}_t\right)K_{\tilde{X}}^{-1}\Psi_{\tilde{X}}^T)\Psi_{\tilde{X}}\tilde{S}_t^{-1}\tilde{m}_t$$

$$= (1 - \gamma_t)\Psi_{\tilde{X}}(I - K_{\tilde{X}}^{-1}\left(K_{\tilde{X}} - \hat{S}_t\right))\tilde{S}_t^{-1}\tilde{m}_t$$

$$= (1 - \gamma_t)\Psi_{\tilde{X}}(I - \left(I - K_{\tilde{X}}^{-1}\hat{S}_t\right))\tilde{S}_t^{-1}\tilde{m}_t$$

$$= \Psi_{\tilde{X}}K_{\tilde{X}}^{-1}\left((1 - \gamma_t)\hat{S}_t\tilde{S}_t^{-1}\tilde{m}_t\right)$$

$$= \Psi_{\tilde{X}}K_{\tilde{X}}^{-1}\hat{m}_t$$

where $\hat{m}_t := (1 - \gamma_t)\hat{S}_t\tilde{S}_t^{-1}\tilde{m}_t$

Thus, the subproblem is a also variational sparse GPR written in the same inducing functions, but with likelihood with modified variance
$$\sigma^2 \leftarrow \frac{\sigma^2}{N\gamma_t}$$

and prior with modified mean and covariance

$$\hat{m}_t \leftarrow (1 - \gamma_t)\hat{S}_t\tilde{S}_t^{-1}\tilde{m}_t$$

$$\hat{S}_t^{-1} \leftarrow (1 - \gamma_t)\tilde{S}_t^{-1} + \gamma_t K_{\tilde{X}}^{-1}$$

## B  Auxiliary Experimental Results

Figure 2: Online learning results of *kin40k*. nMSE evaluated on the held out test set; $N_m = 2048$.

(a) joint 1

(b) joint 2

(c) joint 3

(d) joint 4

(e) joint 5

(f) joint 6

(g) joint 7

Figure 3: Online learning results of *sarcos*. nMSE evaluated on the held out test set; the dash lines and the solid lines denote the results with $N_m = 512$ and $N_m = 2048$, respectively.

(a) joint 1

(b) joint 2

(c) joint 3

(d) joint 4

(e) joint 5

(f) joint 6

(g) joint 7

Figure 4: Online learning results of *KUKA1*. nMSE evaluated on the held out test set; $N_m = 2048$

(a) joint 1

(b) joint 2

(c) joint 3

(d) joint 4

(e) joint 5

(f) joint 6

(g) joint 7

Figure 5: Online learning results of *KUKA2*. nMSE evaluated on the held out test set; $N_m = 2048$