[Reviews · NeurIPS 2016]

Reviewer 1

Summary

This work focuses on learning GP regression models in a streaming setting. This is an important problem and an interesting contribution, but the authors miss a key reference in the field. This is work they should be aware of and compare to.

Qualitative Assessment

The main contribution of this paper is to propose an incremental learning algorithm for variational sparse GP regression based on stochastic mirror ascent of probability densities in reproducing kernel Hilbert space. The majority of GP regression approaches are batch methods that operate on the entire training dataset, precluding the use of datasets that are streaming or too large to fit into memory. As such this is very interesting work. For example, the result presented in 3.2 is a great insight. Section 3 could be made more accessible, but overall it is relatively well explained. A summary algorithm would help the practically minded reader. My main concern is the fact that the authors miss a key reference in this area. I think it is critical they discuss the relation to this work and consider it as a baseline in their experiments. The authors only report results for a fixed number of inducing function. Could they discuss why this is the case irrespective of the data set? How do the methods perform if this number is optimized, let say by cross-validation? Also, it was not clear to me why the proposed selection method of the hyperparameters is a good one (optimal in batch mode). Could the authors comment on the way they would set these in practice and how the performance would look like?

Confidence in this Review

3-Expert (read the paper in detail, know the area, quite certain of my opinion)


Reviewer 2

Summary

This paper proposes an approach for solving variational sparse Gaussian process regression in an on-line fashion. Algorithmically, the main idea is to perform stochastic mirror ascent on the variation lower bound to update the variational distribution of the inducing variables and at the same time do stochastic gradient ascent to update the hyper-parameters.

Qualitative Assessment

Overall, although this paper has some interesting ideas, I find it difficult to read, for example, the motivation is not well stated; the ideas about the dual presentation in RKHS and the subspace parameterization are not well presented (at least not in the sense of mathematical rigorousness); some notations are used before even properly introduced. The contribution of this paper (apply mirror ascent to ELBO?) is somewhat unclear to me; if there's any, it seems to be incremental. Line 19: the merit of GP, as a Bayesian method, is not finding MAP, but offers full posteriors distribution. Line 67: what's \mathcal{X}? Line 78: I find it confusion that your are using both Y (the set form) and y (the vector form), is introducing Y necessary? Line 83: period after equation (3). In general, be consistent of using punctuations after equations. Line 86: Equation (4), none has been said about what \perp means. Line 110: where does p(\tilde{X}) come from? Line 122: another related paper is [1], where the authors re-parameterize the variational lower bound in terms of the summation of the data points. Line 134: it's unclear to me what the sentence "this avoids the issue of ... " mean. Line 145: I don't think in general it includes *all* continuous functions on a compact set. Line 148: k_x is not a vector, but a function in H now. In equation (9), you have yet to define the quadratic form (\phi_x^T\Sigma\phi_x) in H. Line 172: for equation (12), what's q(f_p)? Line 186: equation (15), where do, suddenly, the inner product in L_2 pop up? [1] Gal et al., Distributed Variational inference in Sparse Gaussian process regression and latent variable models

Confidence in this Review

2-Confident (read it all; understood it all reasonably well)


Reviewer 3

Summary

This paper proposes a new and novel approach to sparse Gaussian process regression using a RKHS formulation. In specific the paper describes previous sparse approximation leading to the variational approximation by Titisias. The problem with this approach is that it aims to compress the full non-parametric model into a set of inducing points by formulating an approximate posterior distribution that leads to a lower bound on the marginal log likelihood which requires visiting the whole data-set in order to learn the model. In this paper the authors proposes working in RKHS dual form rather than in its primal. This allows the authors to formulate a slightly different bound compared to the standard variational approach which allows for incremental learning. The methodology section of the paper concludes with how this incremental approach can be implemented and relates to previous methods. The paper ends with a set of experimental results comparing the proposed RKHS approach to the Stochastic Variational GP approach by Hensmann et. al.

Qualitative Assessment

This is a really nice paper, the writing is excellent, clearly explaining what the paper proposes and doing so in a nice "story" that includes previous work and how the proposed approach relate to them. I am impressed how the authors basically manages to explain everything from GPs over sparse approximations to the latest variational approximations across basically two pages. It is refreshing to read a paper which does not hide relationships to previous work and overstates its own proposed solution but lets it be a natural part of the story. The experiments are sufficient for the paper and clearly shows the benefit of the paper. ** Reviewer confidence regarding this review.

Confidence in this Review

2-Confident (read it all; understood it all reasonably well)


Reviewer 4

Summary

The authors introduce the sparse inducing point variational inference scheme for Gaussian processes from the RKHS point of view. This allows for an optimisation algorithm, based on mirror descent, that takes the information geometry of the inducing inputs into account, as well as other variational parameters. This has the possibility of improving the convergence rate of stochastic inference in GPs, and could allow for incremental learning.

Qualitative Assessment

The ideas in this paper are interesting and novel. Analysing variational inference for GPs from the RKHS point of view is an interesting exercise, that is bound to lead to interesting insights. The mirror descent algorithm that follows from it also seems to have the ability to solve the problem of stochastically optimising the inducing inputs, as the authors point out. I have one main issue with the paper regarding the claim that the posterior density has a dual representation as a distribution in an RKHS, with an infinite dimensional mean function and a covariance operator. I am worried that some important technical details are swept under the rug. One good example of a possible issue, is that for many kernels (e.g. the squared exponential) samples of the distribution are not even elements of the RKHS (norm of the sampled function is infinite)! Secondly, integrals are being written w.r.t. what seems to be a Lebesgue measure over the infinite dimensional function f. This measure doesn't exist. Often, results obtained through arguments like these work similarly after a more formal treatment. In conclusion: The paper has very interesting contributions. However, I have strong reservations about the technical aspects mentioned above. If the authors address these points in the rebuttal I may be inclined to change the score. Minor points: The citation of the book "Gaussian processes for machine learning" misses a reference to Chris Williams. Line 52: The SVI paper by Hensman does provide a valid method for optimising the inducing points (in contrast to what is said). The stochastic estimate of the gradient w.r.t. the inducing inputs is unbiased. However, the convergence may be very slow, and they were fixed in the experiments, as mentioned in the paper.

Confidence in this Review

2-Confident (read it all; understood it all reasonably well)


Reviewer 5

Summary

Traditional Gaussian process regression (GPR) scales O(N^3) in complexity and O(N^2) in memory, so researchers often encounter datasets that are too large to work with all at once. Sparse GPR approximates the full GP using a set of M < < N inducing functions (bounded linear operators applied to the data). This requires learning: (1) hyperparameters, (2) parameters for the inducing functions, and (3) statistics for the inducing functions. Most approaches require having the entire dataset in memory in order to learn (1-3), but recently some work has been done with variational batch algorithms to allow for incremental learning. This paper introduces a novel algorithm for variational sparse GPR that leverages a dual formation of the problem in reproducing kernel Hilbert space to learn (1-3) with batches of data in an online setting.

Qualitative Assessment

This is a good contribution on a problem of both academic interest and practical significance.

Confidence in this Review

2-Confident (read it all; understood it all reasonably well)


Reviewer 6

Summary

The author propose a novel variational approximation for Gaussian Process regression, by parametrizing the approximate posterior with a set of inducing functions, rather than inducing points. This approach makes on-line learning more feasible, because the the inducing points in previously used approximations are usually a subset of the data, which would not be available in an on-line setting. The approach is developed by using the dual representation of a GP in a RKHS. The exposition is mostly understandable, and the results are justifying the effort.

Qualitative Assessment

This is a very interesting, well written paper that tackles a relevant topic for the NIPS audience. Minor suggestions for improvement below: - line 86, eqn.4: I assume f_{\tilde{X}} and f_X are the function values at the inducing points and the training data? please define. - line 110: "...whereas p(\tilde{X})..." do you mean p(f_{\tilde{X}) ? - line 126: "...of minimizing (8)..." do you mean maximizing? - lines 149-150: "The mean and the covariance functions...can be equivalently represented as a Gaussian distribution." this sentence made no sense to me. How can a distribution represent two functions? - line 156: "...such that \Sigma \geq 0..." should this be \tilde{\Sigma}? - line 172, eqn. (12): can the derivative of this expression, paricularly of the square root of the difference of determinants, lead to numerical instability when the difference btw. the matrices gets small? Please comment. - line 179: "...without referring to the samples.." this point is so important that it would be helpful for the reader if you wrote down the sample-independent form of the posterior here, so one does not have to piece it together from eqns 10-11 and the explanations that around these equations. line 188: "...with error \epsilon_t..." please define what this error is. line 191, eqn 16: the step from the second to the third line is puzzling, and the reader has to work through a whole paragraph before s/he is referred to the appendix where you explain the step. Please put that reference closer to where you need it. - line 244: "...inducing functions to 512": ho did you come up with this number? what happens if you use less or more? Please also write down the inducing functions. Proofs: - line 348: This derivation requires a number of auxiliary results which are proven afterwards. This is confusing for the reader. Please prove the auxiliary results first. Also, I fould the step from line 3 to 4, and from line 5 to 6 could do with more explanation. - line 349: please explain how you arrive at the first identity. - line 350: please explain how you arrive at the first identity. - line 354: second line of equations. please explain why the numerator can be rewritten like the denominator. - line 354: last line of equations. I think there should be no p(f_X|f_{\tilde{X}) in the numerator. - line 367: what is \bar{\mu}?

Confidence in this Review

2-Confident (read it all; understood it all reasonably well)